# Economic evaluation of Manchester procedure versus sacrospinous hysteropexy: A follow-up analysis of a randomized clinical trial

Sascha F. M. Schulten[1]*, Rosa A. Enklaar[1], Mirjam Weemhoff[2],
Hugo W.F. van Eijndhoven[3], Sanne A.L. van Leijsen[4], Eddy M.M. Adang[5],
Kirsten B. Kluivers[1], for the SAM study group[¶]

1 Department of Obstetrics and Gynecology, Radboud University Medical Center, Nijmegen, the Netherlands, 2 Department of Obstetrics and Gynecology, Zuyderland Medical Center, Heerlen, the Netherlands, 3 Department of Obstetrics and Gynecology, Isala, Zwolle, the Netherlands, 4 Department of Obstetrics & Gynecology, Máxima Medical Center, Veldhoven, the Netherlands, 5 Department for Health Evidence, Radboud University Medical Center, Nijmegen, the Netherlands

¶ Membership of the SAM study group is provided in the Acknowledgment
☯ These authors contributed equally to this work.
* sascha.schulten@radboudumc.nl

## Abstract

### Background

Pelvic organ prolapse is a common condition in females. The reported lifetime risk of undergoing pelvic organ prolapse surgery is estimated to affect up to 20% of women. Recently, a higher level of surgical success after the Manchester procedure has been shown compared to sacrospinous hysteropexy. As the costs in healthcare are rising, it is also important to consider the resources and associated cost implications of the choice between these two procedures. An economic evaluation was conducted to compare the alternative costs and benefits.

### Methods

An economic evaluation alongside a randomized controlled trial (RCT) was performed from a societal and healthcare perspective at 2 years of follow-up according to the intention to treat principle. The RCT was a multicenter, randomized, open label trial, executed in 26 Dutch hospitals. 434 women were randomly assigned to the Manchester procedure or sacrospinous hysteropexy. Direct costing data were obtained from electronic case report forms and Medical Consumption Questionnaires. Indirect costing data were obtained by the Productivity Cost Questionnaire. Quality-adjusted Life Years (QALYs) were calculated from the scores on the Euroqol5D-5L questionnaire. Mean cost differences and their 95% confidence intervals (CI) were calculated.

which permits unrestricted use, distribution, and reproduction in any medium, provided the original author and source are credited.

**Data availability statement:** The dataset is available from: https://doi.org/10.34973/wkz9-c249. The dataset contains the original trial protocol, statistical analysis plan, questionnaire data (regarding iMTA PCQ and MCQ, EQ5D-5L) and case record form data (regarding baseline characteristics and retreatments).

**Funding:** KK was funded by the Netherlands Organisation for Health Research and Development (ZonMw) grant 80-84300-98-83006, https://www.zonmw.nl/. The funder had no role in study design, data collection and analysis, decision to publish or preparation of the manuscript.

**Competing interests:** The authors have declared that no competing interests exist.

## Results

From the societal perspective, the Manchester procedure was significantly less expensive than sacrospinous hysteropexy, with a mean difference of 1458.34 euros (95% CI −2746.16 to −170.52). There was no significant difference in the number of QALYs gained over period of 2 years between the arms: 1.67 QALYs (95% confidence interval (95% CI) 1.63 to 1.71) for the sacrospinous hysteropexy group and 1.68 QALYs (95% CI 1.65 to 1.72) for the Manchester procedure group (p = 0.346).

## Conclusions

During two years of follow-up the Manchester procedure and sacrospinous hysteropexy showed no statistically significant different effectiveness in terms of QALYs gained against significantly higher costs for sacrospinous hysteropexy.

## Introduction

Pelvic organ prolapse (POP) is a common health problem with an estimated lifetime risk for women of 11–20% to undergo surgical correction for POP [1–3]. Worldwide the first choice for surgical correction of uterine prolapse is transvaginal surgery [4]. In the past decade uterus preserving surgical correction of POP is becoming more popular [5–7]. The results of a large randomized clinical trial (RCT) showed positive results after uterus preservation, with lower rates of anatomic prolapse recurrence after sacrospinous hysteropexy versus vaginal hysterectomy with uterosacral ligament suspension after 5 years of follow-up [8,9]. Another uterus-preserving surgical technique is the Manchester procedure. This technique is defined in the international consensus statement as an amputation of the uterine cervix and plication of the uterosacral ligaments extraperitoneally above the remaining cervical stump [10]. The Manchester procedure and the sacrospinous hysteropexy coexist and are largely performed for the same indications. In 2023, a higher level of surgical success was shown for the Manchester procedure compared to sacrospinous hysteropexy in a large multicenter RCT among women with moderate uterine descent. Patients who had undergone the Manchester procedure had a 10% higher chance of success, defined as the combination of absence of recurrent prolapse, recurrent symptoms and/or a reintervention after 2-year follow-up (77.0% versus 87.3%, respectively, p = 0.007) [11].

In the Netherlands, around 15.000 surgeries for POP are performed annually, which indicates a significant burden on healthcare costs [7]. Cost-effectiveness (CE) analyses are needed to provide the most appropriate care with available resources. This is particularly relevant in countries with a publicly funded health care system. It contributes to efficiency and sustainability in healthcare. Now that the Manchester procedure proved to be the most effective procedure in most women who undergo their first vaginal POP operation, the question raises whether an economic evaluation similarly favors the direction of the Manchester procedure. This could be an extra incentive to the need for implementation of this procedure.

In this study we present an economic evaluation that was performed as a prospective secondary analysis alongside the SAcrospinous hysteropexy versus the Manchester procedure (SAM) study, investigating the CE of the Manchester procedure versus the sacrospinous hysteropexy.

## Materials and methods

This economic evaluation was performed alongside the SAM study, a multicenter, randomized, unblinded clinical trial conducted in 26 Dutch hospitals.

The study protocol, statistical analysis plan, CONSORT checklist and study outcomes have been published previously [11]. In short, women were eligible for the trial if aged 18 years or older and undergoing their first surgery for symptomatic POP, including uterine descent but POP quantification (POP-Q) point D was not allowed to be beyond minus 1 cm (POP-Q D ≤ −1 cm). These inclusion criteria imply that approximately 10% of women with the most severe uterine descent could not be included in the SAM study. After written informed consent, women were randomly assigned in a 1:1 ratio to either Manchester procedure or sacrospinous hysteropexy between July 3, 2018, and February 18, 2020.

In sacrospinous hysteropexy, the uterus is suspended unilaterally to the right sacrospinous ligament with two nonabsorbable sutures running through the posterior side of the cervix. The Manchester procedure consists of extraperitoneal plication of the uterosacral ligaments at the posterior side of the uterus and amputation of the cervix. Furthermore, the cardinal ligaments are plicated on the anterior side of the cervix. More details on the surgical techniques are described elsewhere [11]. In the event that a woman experiences symptoms of recurrence, such as a sensation of bulging, a re-intervention may be considered. The decision to proceed with re-intervention depends on the pelvic floor symptoms, anatomy and stage of the recurrent prolapse, as well as the woman's preferences. Re-intervention options include pessary treatment or surgery. In case surgery is needed after sacrospinous hysteropexy or the Manchester procedure, the options include anterior and/or posterior colporrhaphy, perineoplasty, vaginal hysterectomy, sacrospinous hysteropexy or the Manchester procedure, (robot-assisted) sacral colpopexy, and/or tension-free vaginal tape (TVT). Currently, there is no standard protocol on the best treatment in case of recurrence after a sacrospinous hysteropexy or Manchester procedure.

The primary outcome was the composite outcome of success after 2 years of follow-up, defined as the absence of vaginal prolapse beyond the hymen, the absence of bothersome bulge symptoms and the absence of retreatment of recurrent prolapse (pessary or surgery) within 2 years of follow-up. The absence of bulge symptoms was defined as a negative response to the question "Do you usually have a bulge or something falling out that you can see or feel in your vaginal area?" (PFDI-20 POPDI-6 domain, question 3; score 0). All three criteria were required to categorize the primary outcome as 'success'. Follow-up included outpatient visits at 6 weeks, 1 and 2 years after surgery (with prospective completion of the electronic Case Report Form (eCRF) physical exam and assessment of complications and retreatments) and completion of validated patient-reported questionnaires after 3, 6, 9, 12 and 24 months after surgery.

### Economic evaluation

The economic evaluation was performed at 2 years of follow-up according to the intention to treat principle. The 2-year follow-up was chosen because all relevant cost and effect differences manifest themselves in that period. The cost-utility analysis adhered to a societal perspective. Additionally, a healthcare perspective is presented. The Consolidated Health Economic Evaluation Reporting Standards 2022 (CHEERS 2022) checklist was followed (S1 Checklist).

Direct costing data during hospital admission were collected on the eCRF in CastorEDC completed by local research nurses. This included the data on the surgical procedure, hospital admission, visits of healthcare professionals, treatments received and medication used. Direct costing data during follow-up were obtained from the iMTA Medical Consumption Questionnaire (MCQ) [12] and the eCRF on the follow-up visits. Indirect costing data about productivity loss were obtained by means of the iMTA Productivity Cost Questionnaire (PCQ) [13]. CastorEDC was accessed on March 22, 2022 and November 21, 2023 for exporting data. Analyses were performed with Stata version 18 and SPSS version 29.

The unit costs were based on reference prices from 2014 and indexed to 2019 [14] based on the Dutch guideline for healthcare cost analysis [13,15], see Table 1 for cost prices. The friction cost-method was applied following the Dutch guideline for economic evaluation in healthcare [16]. The costs for medication were calculated using standard costs per medication [17]. Costs were expressed as means ±standard error (SE). Mean cost differences and their 95% confidence intervals (CI) were calculated (univariable analysis). To take a potentially skewed distribution into account, as well as possible heteroscedasticity, a generalized linear model (GLM) with a gamma distribution was estimated on the societal and healthcare costs.

Health-related quality of life was based on the patients scores on the Dutch version of the Euroqol5D-5L (EQ-5D-5L), measured at six moments: preoperatively and 3, 6, 9, 12 and 24 months postoperatively. Quality-adjusted life years (QALYs) were calculated over the period of evaluation using the EQ-5D-5L index scores (utilities) multiplied by the consecutive time periods using the trapezium method. The EQ-5D-5L Dutch tariff was used to calculate utilities. QALYs were also estimated using a generalized linear model with gamma distribution.

**Table 1. Cost prices in euros.**

| Resources | Source | Price |
|---|---|---|
| **Healthcare related costs** | | |
| **Intervention** | | |
| Sacrospinous hysteropexy including anterior and/or posterior colporrhaphy | Mean cost price non-academic hospital | 2,458 |
| Manchester procedure including anterior and/or posterior colporrhaphy | Mean cost price non-academic hospital | 2,566 |
| Capio device | Boston Scientific | 280 |
| Capio suture | Boston Scientific | 22 |
| Outpatient clinic visit | Reference price 2014 | 91 |
| **Re-intervention** | | |
| Pessary | Pelvitec.nl | 35 |
| TVT | Mean cost price non-academic hospital | 1,315 |
| Robot assisted surgery | Mean cost price non-academic hospital | 9,307 |
| Vaginal hysterectomy | Mean cost price non-academic hospital | 2,808 |
| Anterior or posterior colporrhaphy | Mean cost price non-academic hospital | 1,762 |
| Manchester procedure | Mean cost price non-academic hospital | 1,816 |
| Perineoplasty | Mean cost price non-academic hospital | 1,762 |
| **Complication treatment** | | |
| Foley catheter | Medireva.nl | 27 |
| Clean intermittent self-catheterization | Mean cost price non-academic hospital | 181 |
| Outpatient clinic hysteroscopy | Mean cost price non-academic hospital | 473 |
| **Other** | | |
| General practitioner visit | Reference price 2014 | 33 |
| Emergency room visit | Reference price 2014 | 259 |
| Ambulance | Reference price 2014 | 515 |
| **Social costs** | | |
| Productivity loss paid work per hour | Reference price 2014 | 32 |
| Productivity loss unpaid work per hour | Reference price 2014 | 14 |
| Travel costs per km | Reference price 2014 | 0.2 |

Prices are presented per 1 unit unless stated otherwise. Reference prices were based on the Dutch guideline for healthcare cost analysis. TVT = tension-free vaginal tape; km = kilometer.

For CE, two data scenarios were performed: a scenario with multiple imputation and a scenario with complete cases only Missing data were assumed to be missing (completely) at random. Multiple imputation by chained equations with predictive mean matching was applied to impute the missing values and ten data sets were created. All available baseline characteristics, operative variables and outcome data were used for imputation. Two perspectives were assessed, a healthcare perspective and a societal perspective (the base-case scenario). Incremental CE ratios (ICERs) were calculated. Costs and effects are typically correlated and hence their correlation should be accounted for. This is done by non-parametric bootstrapping and seemingly unrelated regression (SUR). When resampling cost and effects in pairs, the correlation structure is kept intact when estimating statistical uncertainty. When using SUR, two separate regression models are specified simultaneously; i.e., one for costs and one for effects. In SUR, the correlation between costs and effects is accounted for through correlated error terms. In the complete case scenario no discounting was applied for improving generalizability across countries. The results are plotted in a CE plane, that graphically illustrate the bootstrapped incremental cost-effect pairs. CE Acceptability Curves (CEAC) were estimated that show the probability of the Manchester procedure being cost effective compared to sacrospinous hysteropexy for a willingness-to-pay (WTP) threshold.

### Ethics statement

The SAM study was approved by the medical ethics committee of region Arnhem-Nijmegen (file number: 2017–3443) in accordance with the declaration of Helsinki. In addition, all local boards of directors gave approval to conduct this trial. All participants provided written informed consent prior to inclusion and randomization. This evaluation is part of the SAM study, registered with TrialRegister.nl (NTR 6978).

## Results

In total 434 women were randomly assigned to sacrospinous hysteropexy (n = 217) or Manchester procedure (n = 217). Two patients in each arm were excluded due to prior urinary incontinence surgery, which was an exclusion criterium. In each arm, 215 women remained.

Baseline characteristics in the treatment groups did not differ significantly, see Table 2 [11]. For peri-and postoperative characteristics, see Table 3. Of all women included in the main analysis (n = 430), 91% had completed follow-up data for the composite outcome of success (n = 393). Follow-up data were complete for 84% of QALYs (n = 363), for 72% of total healthcare costs (n = 1653, 823 in control group and 830 in intervention group), and for 58% of total societal costs (n = 1335, 671 in control group and 664 in intervention group). See S2 Fig for patterns of missing data on EQ-5D-5L and S3 Fig for patterns of missing data on MCQ and PCQ at different measurements.

### Costs

From the societal perspective, the Manchester procedure was significantly less expensive than sacrospinous hysteropexy, with a mean difference of 1458.34 euros (95% CI −2746.16 to −170.52), as shown in Table 4. Discounting did not have a significant influence on these results.

From the healthcare perspective, the Manchester procedure was less expensive than sacrospinous hysteropexy, but the difference was not statistically significant (mean difference 310.85 euros; 95% CI −767.06 to 145.36), as shown in Table 4.

### Complete case scenario

From the societal perspective, the Manchester procedure was significantly less expensive in the complete case scenario than sacrospinous hysteropexy (7996.13 versus 9526.79 euros, mean difference 1530.65 euros; 95% CI −2952.53 to −108.77).

**Table 2. Baseline characteristics by treatment group.**

| Characteristics | Sacrospinous hysteropexy (n=215) | Manchester procedure (n=215) |
|---|---|---|
| Age in years, median (Q1-Q3) | 61 (55-69) | 63 (56-70) |
| Ethnicity | | |
| Caucasian | 185 (86.0%) | 183 (85.1%) |
| Highest educational level | | |
| Primary or secondary school | 82 (38.1%) | 92 (42.8%) |
| High school | 74 (34.4%) | 77 (35.8%) |
| Bachelor, master or academic degree | 55 (25.6%) | 45 (20.9%) |
| Comorbidity | | |
| Cardiovascular disease | 50 (23.3%) | 51 (23.7%) |
| Diabetes mellitus | 11 (5.1%) | 11 (5.1%) |
| Respiratory disease | 25 (11.6%) | 17 (7.9%) |
| Current smoker | 15 (7.0%) | 15 (7.0%) |
| Postmenopausal status | 172 (80.0) | 174 (80.9) |
| Body mass index in kg/m$^2$, median (Q1-Q3) | 25.4 (23.3-28.0) | 25.2 (23.2-28.6) |
| Overall POP-Q stage [a] | | |
| 2 | 87 (40.5%) | 106 (49.3%) |
| 3 | 128 (59.5%) | 108 (50.2%) |
| 4 | 0 (0%) | 1 (0.5%) |

Data presented as numbers (percentages) unless stated otherwise. POP-Q=pelvic organ prolapse quantification.
[a] Stage POP-Q: stage 2: most distal prolapse is between 1cm above and 1cm beyond hymen; stage 3: most distal prolapse is prolapsed >1cm beyond hymen but no further than 2cm less than total vaginal length; stage 4: total prolapse.

From the healthcare perspective, the Manchester procedure was less expensive in the complete case scenario than sacrospinous hysteropexy, but the difference was not statistically significant (4151.44 versus 4525.22 euros, mean difference 373.78 euros; 95% CI −849.79 to −102.22).

### Effects

There was no significant difference in the number of quality-adjusted life years (QALYs) gained over period of 24 months between the two groups: 1.67 QALYs (95% confidence interval (95% CI) 1.63 to 1.71) for the sacrospinous hysteropexy group and 1.68 QALYs (95% CI 1.65 to 1.72) for the Manchester procedure group (p=0.346), see Table 5 for the utilities per measuring moment. Discounting did not have a significant influence on these results.

### Incremental cost-effectiveness ratio

Fig 1 shows the CEACs from the societal and healthcare perspective. The probability that the Manchester procedure is cost effective, compared to sacrospinous hysteropexy, decreases as the WTP increases.

S4 and S5 Fig show the CE planes of the societal perspective and health care perspective, respectively, presenting the bootstrapped incremental CE ratios. In the CE plane for societal perspective, 65% of the points are in the South-East quadrant. In the CE plane for health care perspective, 51% of the points are in the South-East quadrant. It shows that sacrospinous hysteropexy and the Manchester procedure have comparable effectiveness expressed as QALYs gained, but sacrospinous hysteropexy is more expensive.

**Table 3. Peri and postoperative characteristics.**

| Characteristics | Sacrospinous hysteropexy (n=207) | Manchester procedure (n=213) |
|---|---|---|
| Operating time, min (IQR) | 63 (48-81) | 62 (48-80) |
| Estimated blood loss, ml (IQR) | 50 (50-100) | 100 (50-137.5) |
| Type of SSH procedure | | |
| With suture capturing device[a] | 114/200 (57.0%) | N/A |
| **Adverse events** | | |
| Urinary retention with treatment: | 37 (18.2%) | 25 (12.3%) |
| Foley catheter | 16 (45.7%) | 9 (40.9%) |
| clean intermittent self catheterization | 24 (68.6%) | 15 (65.2%) |
| Foley catheter and clean intermittent self-catheterization | 7 (19.4%) | 2 (8.7%) |
| Re-operation reason: | 3 (1.4%) | 3 (1.4%) |
| (delayed) haemorrhage needing surgery | 1 (0.5%) | 2 (0.9%) |
| suture removal | 3 (1.4%) | NA |
| Re-hospitalization &; for reason: | 7 (3.3%) | 4 (1.9%) |
| suture removal | 2 (1.0%) | 0 (0%) |
| urinary retention | 4 (1.9%) | 0 (0%) |
| infection | 3 (1.4%) | 1 (0.5%) |
| delayed hematoma (needing surgery) | 0 (0%) | 1 (0.5%) |
| constipation | 0 (0.0%) | 1 (0.5%) |
| **Repeat surgery for prolapse recurrence** | | |
| in operated compartment | 9/207 (4.3%) | 0/212 (0.0%) |
| in non-operated compartment | 2/154 (1.3%) | 4/144 (2.8%) |
| **Primary outcome: composite outcome of success[a]** | 151/196 (77.0%) | 172/197 (87.3%) |
| Absence of POP beyond hymen | 170/196 (86.7%) | 187/195 (95.9%) |
| Absence of bulge symptoms | 179/200 (89.5%) | 183/201 (91.0%) |
| Absence of reintervention | 187/199 (94.0%) | 195/201 (97.0%) |

Data presented as numbers (percentages) unless stated otherwise. IQR; interquartile range. N/A: not applicable. [a] in 46.5% of cases, an open procedure was performed. [a] Composite outcome of success defined as the absence of POP beyond the hymen in any compartment, and the absence of bulge symptoms and the absence of reoperation or pessary treatment for POP.

## Discussion

We assessed the CE of the Manchester procedure versus sacrospinous hysteropexy. From a societal perspective, during two years of follow-up, we found no statistically significant evidence of difference for effectiveness expressed in QALYs against significantly higher costs for sacrospinous hysteropexy. Direct (hospital) costs were similar. These CE results add to the results of the SAM study, which showed inferiority of sacrospinous hysteropexy versus the Manchester procedure for a composite outcome of success (based on anatomy, complaints and re-intervention) [11].

A recent study on the national costs for POP surgery in the United States showed an annual cost of $1.523 billion per year. The median cost per procedure was $8,958 in 2018 [18]. This burden underscores the importance of CE studies. However, few CE studies are available on surgical treatment for POP. The study by Wang et al. used a Markov model with a time horizon of 5 years for estimating CE of vaginal apical suspension, laparoscopic sacrocolpopexy, and robotic sacrocolpopexy compared to expectant management [19]. They showed that surgical intervention is always cost effective compared to expectant management. Interestingly, they showed that the vaginal approaches are more cost effective

**Table 4. Mean costs in euros, indexed at 2019 prices.**

| | Sacrospinous hysteropexy (n=215) | Manchester procedure (n=215) | Mean difference (95%CI) |
|---|---|---|---|
| **Total direct (healthcare) costs** | 4,737 (179) | 4,426 (148) | −311(−767–145) |
| Primary surgery | 2,628 (8) | 2,567 (8) | −61 (−83 to −40) |
| Total MCQ costs | 1,831 (142) | 1,737 (139) | −94 (−483–295) |
| General practitioner | 183 (12) | 144 (8) | −39 (−67 to −12) |
| Paramedical care | 339 (38) | 328 (37) | −11 (−114–93) |
| Outpatient care | 411 (25) | 349 (22) | −63 (−128–2) |
| Medication | 81 (7) | 63 (6) | −18 (−36–1) |
| Home care | 275 (89) | 324 (89) | 48 (−199–196) |
| Hospital admission | 485 (54) | 450 (64) | −35 (−200–130) |
| Emergency department visit | 52 (11) | 68 (13) | 16 (−18–50) |
| Ambulance | 34 (15) | 37 (11) | 2 (−35–39) |
| Extra costs [a] | 278 (59)[b] | 122 (26)[b] | −156 (−282 to −30) |
| **Indirect non-medical costs** | | | |
| Total PCQ costs | 5,110 (429) | 3,966 (376) | −1,145 (−2263 to −26) |
| Short term absenteeism | 2,225 (264) | 1,735 (212) | −489 (−1,154–174) |
| Long term absenteeism | 813 (197) | 826 (188) | 13 (−521–547) |
| Presenteeism | 518 (84) | 453 (74) | −95 (−316–125) |
| Unpaid productivity loss | 1,527 (172) | 951 (133) | −576 (−1,001 to −150) |
| Total travel costs | 21 (1) | 19 (1) | −3 (−6–1) |
| **Societal costs** | 9,869 (506) | 8,410 (419) | −1,458 (−2746 to −171) |

Costs (€) for the multiple imputation scenario were estimated with a generalized linear model with gamma distribution, expressed as mean ± standard error. 95%CI: 95% confidence interval; MCQ: medical consumption questionnaire; PCQ: productivity cost questionnaire.

[a] Extra costs: costs for re-operation for recurrence or complication, pessary, self-catheterization and/or Foley catheter.

[b] Costs were incurred for sacrospinous hysteropexy n=50 and Manchester procedure n=36.

**Table 5. Mean utility scores and QALYs per measuring moment.**

| Measuring moment | Sacrospinous hysteropexy (n=215) | Manchester procedure (n=215) | Sig. |
|---|---|---|---|
| Pre-operative | 0.80±0.19 | 0.80±0.18 | 0.96 |
| 12 weeks postoperative | 0.88±0.15 | 0.90±0.14 | 0.18 |
| 6 months postoperative | 0.89±0.14 | 0.89±0.15 | 0.68 |
| 9 months postoperative | 0.89±0.15 | 0.88±0.20 | 0.52 |
| 12 months postoperative | 0.89±0.15 | 0.89±0.16 | 0.78 |
| 24 months postoperative | 0.88±0.17 | 0.90±0.17 | 0.20 |
| QALY 12 months postoperative | 0.85±0.13 | 0.84±0.15 | 0.81 |
| QALY 24 months postoperative | 0.88±0.15 | 0.90±0.15 | 0.21 |
| QALY total | 1.69±0.29 | 1.70±0.30 | 0.42 |

Mean utility scores and QALYs were calculated from the Euroqol5D-5L questionnaire ± standard deviation. Statistical significance was calculated with Student's T-test.

for patients with shorter life expectancy. The study by Wallace et al. used a Markov model to estimate CE comparing uterus preserving strategies with vaginal hysterectomy strategies [20]. The recurrence rates for sacrospinous hysteropexy were based on the 5-year follow-up results of the SAVE-U study [8]. The authors concluded that vaginal hysterectomy

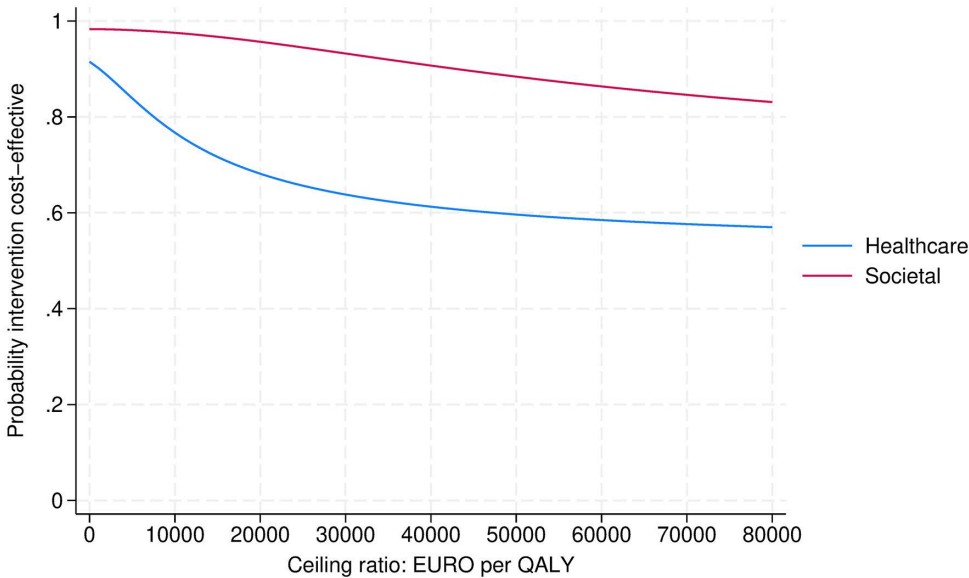

**Fig 1. CEAC societal perspective and health care perspective.**

strategies are only cost effective in case prolapse recurrence rates are at least 16% after hysteropexy and 15% would need repeat surgery. In the SAM study, the thresholds for repeat surgery were not reached within two years postoperatively, and it is unlikely that they will be reached at the five-year follow-up. Data for this extended follow-up are currently being collected.

A study by Chang et al. showed that vaginal hysterectomy compared to sacrospinous hysteropexy has higher costs, but is a cost-effective approach for prevention of endometrial cancer by reducing subsequent major surgery [21]. Abovementioned studies had longer time horizons than our study, which could put results in a different perspective.

Interestingly, the sacrospinous hysteropexy was significantly more expensive from a societal perspective only. This implies that the difference in costs is primarily caused by paid and unpaid productivity losses post-surgery (i.e., indirect not-medical costs). Most previous CE studies on POP are performed using models from a healthcare perspective, i.e., without this information on productivity loss. Our study shows that productivity loss should be considered, as significant differences can exist. At baseline, PCQ costs were higher for the sacrospinous hysteropexy group, but this difference was not statistically significant (mean difference −62.75 95%CI −406.28 to 280.79). At all the follow-up moments the PCQ costs were higher in the sacrospinous hysteropexy group. Both the costs and the difference in costs were peaking at 3 months follow-up, but the difference in costs at this point was not statistically significant (sacrospinous hysteropexy €2381.37 versus the Manchester procedure €2899.02, difference −517.65, 95%CI −1339.76 to 304.46).

The healthcare perspective, which is part of the societal perspective, showed a difference in costs in favor of the Manchester procedure, but this difference was not statistically significant. We had expected more hospital costs for sacrospinous hysteropexy as a disposable device (€325 per procedure) was used in 57% of procedures. However, the cost price for a Manchester procedure is higher, this might be due to a cost price calculation in which a longer operating time for Manchester procedure is used in some hospitals. This compensates for the costs of the device used for sacrospinous hysteropexy. No difference in operating time was however found. Moreover, we anticipated that repeat surgery for prolapse would result in more pronounced cost differences; however, repeat surgery alone was not a decisive factor."

In the present study, effectiveness was defined in terms of QALYs gained as is common in CE studies. QALYs gained were similar between groups. The primary outcome in the SAM study was a composite outcome of surgical success,

which showed more favorable results after the Manchester procedure. Recently it has been advised to primarily define surgical success after POP surgery as the absence of bothersome patient bulge symptoms and absence of retreatment (and thereby disregard anatomy as a relevant outcome) [22]. In that analysis, there were similar outcomes as in the QALY assessment. There were furthermore no differences between groups in either bothersome bulge symptoms or patient's global impression of improvement at two years of follow-up in the SAM study.

This is the first CE study on the Manchester procedure versus sacrospinous hysteropexy. The major strength of this study is that it was performed alongside a multicenter, randomized unblinded clinical trial. Furthermore, the CE analysis was performed following the Dutch guideline on economic evaluations in healthcare [23].

A limitation of our study was the inability to calculate the indirect costs for all participants as some PCQ's were missing: in the sacrospinous hysteropexy group 19 (8.8%) women were lost to follow-up and 20 (9.3%) women in the Manchester procedure group. Women who were lost to follow-up were older and had more cardiovascular morbidity [11]. Another limitation is that additional measurements of the MCQ and PCQ during the interval 12–24 months would have provided more detail but were not available. However, logically, costs regarding the primary surgery (including revalidation and treatment of complications) are considered to be incurred within the first year after surgery. Furthermore, data on repeat surgery for recurrences, which may be incurred in the second year as well, are less likely to be prone for recall bias.

### Implications for clinical practice

Even in case of partial replacement of sacrospinous hysteropexy by Manchester procedure in the Netherlands, this would imply a reduction in costs. Our analysis shows similar effectiveness in terms of quality of life, and effectiveness is thereby no justification for the additional costs for sacrospinous hysteropexy.

It is known that the doctor's preference is mainly based on the gynecologist's own experience and background [24]. Before publication of the results of the SAM study, lack of information on the comparison of Manchester versus sacrospinous hysteropexy hampered evidence-based decision making, causing practice pattern variation. The publications of the SAM study and this CE analysis now point out clear differences between the operations. Internationally, gynecologists are hopefully encouraged to further implement the Manchester procedure into their daily practice.

### Conclusion

This study assessed the CE of the Manchester procedure versus sacrospinous hysteropexy. During two years of follow-up, sacrospinous hysteropexy and the Manchester procedure showed similar effectiveness in terms of QALYs against significantly higher costs for sacrospinous hysteropexy.

### Supporting information

**S1 Checklist. CHEERS 2022 Checklist.**
(PDF)

**S1 Fig. Pattern of missing data on EQ-5D-5L; Percent of cases per missing value pattern.**
(PDF)

**S2 Fig. Pattern of missing data on PCQ and MCQ costs; Percent of cases per missing value pattern.**
(PDF)

**S3 Fig. CE plane societal perspective.**
(PDF)

**S4 Fig. CE plane healthcare perspective.**
(PDF)

## Acknowledgments

We thank the members of the SAM study group for their contribution in data acquisition: Jeroen van Bavel (Amphia Hospital, Breda, the Netherlands); Anna C. Verkleij-Hagoort (St Antonius Hospital, Nieuwegein, the Netherlands); Joggem Veen (Máxima Medical Centre, Veldhoven, The Netherlands); Diana Massop-Helmink (Medisch Spectrum Twente, Enschede, The Netherlands); Marko Sikkema (ZGT, Almelo, The Netherlands); Charlotte Lenselink (Deventer Ziekenhuis, Deventer, The Netherlands); Pieternel Steures (Jeroen Bosch Hospital, s Hertogenbosch, The Netherlands); Chantal Wingen (Laurentius Hospital, Roermond, The Netherlands); Kim J.B. Notten (Radboud university medical center, Nijmegen, The Netherlands); Deliana van Rumpt-van de Geest (Reinier de Graaf Hospital, Delft, The Netherlands); Jorik Vellekoop (Zuyderland Medical Center, Heerlen, The Netherlands); Maria K. Engberts (Isala, Zwolle, The Netherlands); Anne Damoiseaux (Catharina Hospital, Eindhoven, The Netherlands); Jackie Stoutjesdijk (Canisius Wilhelmina Hospital, Nijmegen, The Netherlands); Ronald J.C. Mouw (Rijnstate Hospital, Arnhem, The Netherlands); Marinus van der Ploeg (Martini Hospital, Groningen, The Netherlands); Iris van Gestel (Viecuri Hospital, Venlo, The Netherlands); Astrid Vollebregt (Spaarne Gasthuis, Hoofddorp, The Netherlands); Jelle Stekelenburg (Medical Center Leeuwarden, Leeuwarden, The Netherlands); Wilbert Spaans (Maastricht University Medical Center, Maastricht, The Netherlands); Stella Tiersma (Amsterdam UMC location Vrije Universiteit Amsterdam, Amsterdam, The Netherlands); Wenche Klerkx (St Antonius Hospital, Nieuwegein, The Netherlands); Leonie Speksnijder (Amphia Hospital, Breda, The Netherlands).

## Author contributions

**Conceptualization:** Sascha FM Schulten, Rosa A Enklaar, Mirjam Weemhoff, Hugo WF van Eijndhoven, Sanne AL van Leijsen, Eddy MM Adang, Kirsten B Kluivers.

**Data curation:** Sascha FM Schulten, Rosa A Enklaar, Eddy MM Adang.

**Formal analysis:** Sascha FM Schulten, Eddy MM Adang.

**Funding acquisition:** Mirjam Weemhoff, Hugo WF van Eijndhoven, Sanne AL van Leijsen, Kirsten B Kluivers.

**Investigation:** Sascha FM Schulten, Mirjam Weemhoff, Hugo WF van Eijndhoven, Sanne AL van Leijsen, Kirsten B Kluivers.

**Methodology:** Sascha FM Schulten, Eddy MM Adang.

**Project administration:** Sascha FM Schulten, Rosa A Enklaar, Kirsten B Kluivers.

**Resources:** Sascha FM Schulten, Rosa A Enklaar, Mirjam Weemhoff, Hugo WF van Eijndhoven, Sanne AL van Leijsen, Eddy MM Adang, Kirsten B Kluivers.

**Software:** Eddy MM Adang.

**Supervision:** Sascha FM Schulten, Eddy MM Adang, Kirsten B Kluivers.

**Validation:** Sascha FM Schulten, Eddy MM Adang, Kirsten B Kluivers.

**Visualization:** Sascha FM Schulten, Eddy MM Adang.

**Writing – original draft:** Sascha FM Schulten.

**Writing – review & editing:** Sascha FM Schulten, Rosa A Enklaar, Mirjam Weemhoff, Hugo WF van Eijndhoven, Sanne AL van Leijsen, Eddy MM Adang, Kirsten B Kluivers.

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
