## [Decision Letter · Decision Letter 0]

29 Jan 2025

Dear Dr. Schulten,

Thank you for submitting your manuscript to PLOS ONE. After careful consideration, we feel that it has merit but does not fully meet PLOS ONE’s publication criteria as it currently stands. Therefore, we invite you to submit a revised version of the manuscript that addresses the points raised during the review process.

**ACADEMIC EDITOR: Please respond to all reviewer comments**
**1. Clarify the type of study in title to avoid confusion as it is a follow up analysis of a RCT**

We look forward to receiving your revised manuscript.

Kind regards,

Ahmed Mohamed Maged, MD

Academic Editor

PLOS ONE

2. In the online submission form you indicate that your data is not available for proprietary reasons and have provided a contact point for accessing this data. Please note that your current contact point is a co-author on this manuscript. According to our Data Policy, the contact point must not be an author on the manuscript and must be an institutional contact, ideally not an individual. Please revise your data statement to a non-author institutional point of contact, such as a data access or ethics committee, and send this to us via return email. Please also include contact information for the third party organization, and please include the full citation of where the data can be found.

Reviewers' comments:

Reviewer's Responses to Questions

**Comments to the Author**

1. Is the manuscript technically sound, and do the data support the conclusions?

Reviewer #1: Yes

2. Has the statistical analysis been performed appropriately and rigorously?

Reviewer #1: Yes

3. Have the authors made all data underlying the findings in their manuscript fully available?

Reviewer #1: Yes

4. Is the manuscript presented in an intelligible fashion and written in standard English?

Reviewer #1: Yes

Reviewer #1: Thank you for the opportunity to read this well conducted and interesting article. Broadly, I think this is a strong article, but more methodological detail is needed. I should note that as a health economist focused on conducting economic evaluations in the UK, I am not familiar with the associated Dutch guidelines – please consider that in relation to my feedback.

I structure my feedback with starting with what I consider the most important points, leading to more minor ones..

Methodological points:

• I feel strongly that the Consolidated Health Economic Evaluation Reporting Standards 2022 (CHEERS 2022) checklist should be completed and followed: https://www.equator-network.org/reporting-guidelines/cheers/

• More detail is needed around missing data: for example, how many imputations were used in the multiple imputation? What were the patterns of missing data? Please review Faria et al 2014 and follow its suggestions: https://pubmed.ncbi.nlm.nih.gov/25069632/

At minimum, I would like to see some plots of missing data in supplementary materials.

• How did you combine bootstrapping with the multiple imputation? Did you bootstrap from each imputed data-set?

• Regression models: did you control for any other variables? Was there any stratification in randomisation of the RCT? If there was, these variables should be included in your regression models. Did you adjust for site? Generally speaking, I would try to follow the statistical approach adopted in the RCT as far as possible – they may used a mixed effects model, which may preclude a SUR model.

• L122-124 – what valuation set was used to turn EQ-5D scores into utilities?

• Was discounting used for the second year of costs and effects? (May depend on the Dutch guidelines)

Supplementary tables with finer detail of resource and cost break down is needed – more detail is needed than just table 4 – perhaps one table, with a row for each “unit cost” resource, comparing arms. Should be repeated for costs.

Precision: Table 4 and more generally: I am very unconvinced that clarity is provided by including cents – why not round costs to the nearest euro? Could you use a thousands separator?

Figures 1-4: I am unconvinced you need both the CE planes and CEACs – why not move the CE-planes to supplementary materials? Also, I think it is a waste of space to have separate figures for the CEACs – why not one plot, with two lines?

L213: “the procedures showed equal effectiveness expressed in QALYs” – I would advise caution with this statement – I think some statisticians would argue that you have not done a non-inferiority study on comparing procedures, so can’t claim equal effectiveness – rather, you have found no statistically significant evidence of difference – the arguments are not reversible. I tend to follow Claxton 1999 (https://doi.org/10.1016/s0167-6296(98)00039-3) on this point, and would focus on the point estimates and the certainty of findings.

More minor points:

L21, Abstract: “Common condition”– include some numbers to speak to this?

L23, Abstract: “As the costs in healthcare are rising, the question raises whether an economic evaluation likewise points out in the direction of the Manchester procedure.” – I found this hard to parse – perhaps something like: “It is also important to consider the resource and associated cost implications of the choice between these two procedures – we conduct an economic evaluation to compare the alternative costs and benefits.”

L61/62: I’m unconvinced the main point is that healthcare costs are rising (not that I disagree) – I would just make the point you want to do the best with available resources – as argued in L63 – perhaps particularly relevant if the healthcare system is publicly funded.

L69: First mention of “SAM study” - should be spelt out/introduced.

Table 1: I find the “units” column unhelpful – I would suggest absorbing the related detail in the resource label, or the unit price – e.g. for the latter, in terms of travel costs – “0.19/km” etc

Table 1: The unit costs introduce the idea of “re-intervention” and complications – these should be motivated/covered when the procedures are introduced/perhaps in the background. For example, if randomised to the Manchester procedure and a re-intervention is necessary, which of the Manchester procedure or Sacrospinous would be used?

Table 5: It would seem natural to extend the table with QALYs at 1 year, 2 years and in total.

L192, L196: To convey some of the uncertainty etc, would it be useful to report the percentage of points falling in the South-East (more-effective, less costly) quadrant of the CE plane?

L218-233: You assert that the important costs and consequences are captured in two years, though others have used models, which I assume have a longer time horizon than 2 years – please add a sentence or two to address this – what was the time horizon in these models? Why didn’t they stop at two years? Etc More broadly, in your literature overview, I think it would be useful to capture the time horizon of these models – is there any “correlation”/link between the time horizon and positive/negative CE findings?

Very minor points:

L65: Perhaps replace: “likewise points out in the direction of” -> “similarly favours”

L75: Replace “in case” with “if”?

L103: eCRF needs spelling at first use.

L109: I think pluralise “unit cost” – more than one unit?

Table 1: TVT needs spelling out

L121: I believe more standard for EQ5D to be capitalised as “EQ-5D-5L”, as per the EuroQol website: https://euroqol.org/information-and-support/euroqol-instruments/eq-5d-5l/

L139: It seems odd to me that data extraction from CastorEDC are mentioned here – why not move this sentence to L102-108 etc?

Table 2: You include variables that are not baseline characteristics (e.g. primary outcome and reintervention etc) – please update the caption. Would some of these variables sit better in Table 3?

L200: Should “if” be “as”?

**Do you want your identity to be public for this peer review?** For information about this choice, including consent withdrawal, please see our Privacy Policy

Reviewer #1: No

---

## [Author Response · Author response to Decision Letter 1]

11 Mar 2025

Dear editor: we have revised our data statement to a non-author institutional point of contact: frontoffice.verlgyn@radboudumc.nl.

Kind regards,

Sascha Schulten

---

## [Editor Report · Decision Letter 1]

12 Mar 2025

PONE-D-24-40477R1

Economic evaluation of Manchester procedure versus sacrospinous hysteropexy: secondary outcome analysis of a randomized clinical trial

PLOS ONE

Dear Dr. Schulten,

Thank you for submitting your manuscript to PLOS ONE. After careful consideration, we have decided that your manuscript does not meet our criteria for publication and must therefore be rejected.

Specifically:

**ACADEMIC EDITOR:**

The authors failed to respond to all reviewers and editors comments

I am sorry that we cannot be more positive on this occasion, but hope that you appreciate the reasons for this decision.

Kind regards,

Ahmed Mohamed Maged, MD

Academic Editor

PLOS ONE

- - - - -

---

## [Author Response · Author response to Decision Letter 2]

26 May 2025

Dear prof Maged,

We are deeply grateful for the opportunity to revise our manuscript again after rejection. We continue to believe that PLOS ONE is the most appropriate journal for the publication of our work.

As the rejection was due to our inability to fully address all of the reviewers' comments, we have now have carefully addressed all comments from the reviewers and the editor. Special consideration was given to the adjustment of responses to the comments of the academic editor, to the reviewer’s comments numbers 7, 17 and 18 (see response letter).

Additionally, the lay-out of the manuscript was adjusted to the journal requirements and the DOI for our dataset is now available and can be accessed openly upon publication at the following link: https://doi.org/10.34973/wkz9-c249. For review purposes, the data is already accessible at the following link: https://data.ru.nl/login/reviewer-37570231/XRGPACSTMX33G7H56G64HEXLZL2KJYOE2MDH6BA.

We sincerely hope that you will reconsider our manuscript for publication.

Kind regards,

Sascha Schulten

---

## [Decision Letter · Decision Letter 2]

11 Aug 2025

Dear Dr. Schulten,

Thank you for submitting your manuscript to PLOS ONE. After careful consideration, we feel that it has merit but does not fully meet PLOS ONE’s publication criteria as it currently stands. Therefore, we invite you to submit a revised version of the manuscript that addresses the points raised during the review process.

We look forward to receiving your revised manuscript.

Kind regards,

Ozan Karadeniz

Academic Editor

PLOS ONE

Journal Requirements:

Additional Editor Comments (if provided):

Thank you for submitting your manuscript to PLOS ONE. After careful consideration, we feel that it has merit but does not fully meet PLOS ONE’s publication criteria as it currently stands. Therefore, we invite you to submit a revised version of the manuscript that addresses the points raised during the review process.

Reviewers' comments:

Reviewer's Responses to Questions

**Comments to the Author**

Reviewer #2: (No Response)

Reviewer #3: All comments have been addressed

2. Is the manuscript technically sound, and do the data support the conclusions?

Reviewer #2: Partly

Reviewer #3: Yes

3. Has the statistical analysis been performed appropriately and rigorously?

Reviewer #2: Yes

Reviewer #3: Yes

4. Have the authors made all data underlying the findings in their manuscript fully available?

Reviewer #2: Yes

Reviewer #3: Yes

5. Is the manuscript presented in an intelligible fashion and written in standard English?

Reviewer #2: Yes

Reviewer #3: Yes

Reviewer #2: Positives :

The title adequately describes the study

The Economic evaluation of Manchester procedure versus sacrospinous hysteropexy" is a comprehensive assessment comparing the alternative costs and benefits of different healthcare procedures

The CHEERS 2022 checklist was followed

Methods used to asses direct costs , indirect costs and QALYs are all standard

Conclusions reached are representative barring a few concerns

Negatives

The 2-year follow-up is too short to reflect long term societal costs based on the definition of failure in this paper [mass symptoms which requires pessary/ surgical intervention]. As all mass symptoms do not require surgery and the progressive nature of prolapse which can worsen over time

Longer follow up of atleast 5 years is often needed to capture ongoing treatment costs, disease progression, and long-term economic consequences as stated in the conclusion

Kindly explain how all relevant cost and effect differences were calculated if [Line 96-97] there is no reliable data available regarding the number of women who undergo specific procedures as a second intervention how can costs be calculated

To conclude [ line 233-34]that the recurrence rates are lower than other comparative studies 2yr vs 5 year follow up is erroneous.

Reviewer #3: The present issue places the consequence of the SAM study, which concluded that sacrospinous hysteropexy is inferior to the Manchester procedure.

Through the present articles, sacrospinous hysteropexy is not entirely inferior to Manchester procedure, taking into consideration about the QALYs.

Actually, almost half of participants belong to the POP-Q 2nd degree. Strictly speaking, those patients tend to be cured even by the use of Pessary, the cheapest procedure. However, this time the highlight is the comparison between sacrospinous hysteropexy and Manchester procedure and not focused on the other procedures.

So far as this second revised articles, there are no suspicious points for me. I really respect your hard works.

**Do you want your identity to be public for this peer review?** For information about this choice, including consent withdrawal, please see our Privacy Policy

Reviewer #2: **Yes: ** Lilly Varghese

Reviewer #3: No

---

## [Author Response · Author response to Decision Letter 3]

18 Aug 2025

Response to reviewers

The reference list was reviewed to identify retracted publications; no such papers were found. Several references were updated:

- Reference 14 was revised to conform to the PLoS Vancouver citation style:

Consumer prices; price index 2015=100 [Internet]. CBS. 2022 [cited 2022 Mar 9]. Available from: https://www.cbs.nl/en-gb/figures/detail/83131ENG

- Reference 16 was updated with the most recent version of the source and a hyperlink was added:

Hakkaart-van Roijen L, Peeters S, Kanters T, van Baal P, Brouwer W, Drost R, et al. Kostenhandleiding voor economische evaluaties in de gezondheidszorg [Report]. Diemen: Zorginstituut Nederland; 2024 [cited 2022 Mar 9]. Available from: https://www.zorginstituutnederland.nl/over-ons/publicaties/publicatie/2024/01/16/richtlijn-voor-het-uitvoeren-van-economische-evaluaties-in-de-gezondheidszorg

- Reference 17 was corrected to align with the PLoS Vancouver style:

Medicijnkosten.nl [Internet]. Zorginstituut Nederland (ZIN). 2023 [cited 2022 May 23]. Available from: https://www.medicijnkosten.nl/

- Reference 23 was updated to reflect the correct publication details:

Zorginstituut Nederland. Richtlijn voor het uitvoeren van economische evaluaties in de gezondheidszorg [Internet]. Diemen: Zorginstituut Nederland; 2016 [cited 2020 Aug 25]. Available from: https://www.zorginstituutnederland.nl/documenten/2024/01/16/richtlijn-voor-het-uitvoeren-van-economische-evaluaties-in-de-gezondheidszorg*

Editor Comments:

Thank you for submitting your manuscript to PLOS ONE. After careful consideration, we feel that it has merit but does not fully meet PLOS ONE’s publication criteria as it currently stands. Therefore, we invite you to submit a revised version of the manuscript that addresses the points raised during the review process.

Review comments

Reviewer #2

Positives:

The title adequately describes the study

The Economic evaluation of Manchester procedure versus sacrospinous hysteropexy" is a comprehensive assessment comparing the alternative costs and benefits of different healthcare procedures

The CHEERS 2022 checklist was followed

Methods used to asses direct costs , indirect costs and QALYs are all standard

Conclusions reached are representative barring a few concerns

Negatives

1. The 2-year follow-up is too short to reflect long term societal costs based on the definition of failure in this paper [mass symptoms which requires pessary/ surgical intervention]. As all mass symptoms do not require surgery and the progressive nature of prolapse which can worsen over time. Longer follow up of at least 5 years is often needed to capture ongoing treatment costs, disease progression, and long-term economic consequences as stated in the conclusion.

Thank you for your review of our paper and your valuable comments.

Balancing feasibility and ambition, we believe that the follow-up period in this study is adequate and allows us to present these findings as relevant and meaningful data. As a cost-effectiveness analysis is inherently incremental, it focuses on differences in costs and QALYs that are most relevant. Within the first two years after surgery, cost differences—particularly those related to productivity—are most pronounced. Given the decreasing incremental cost differences between the procedures over time, we are confident that this timeframe captures the most critical period of cost development.

After two years, both groups tend to converge in terms of healthcare utilization and, indirectly, productivity-related costs. This convergence is also reflected in quality of life outcomes, which are largely comparable between groups at the two-year mark. The observed cost difference of 1458 euros per patient suggests that these differences are likely to remain meaningful, even with a longer follow-up period. While individual patients may experience long-term consequences of surgical treatment for prolapse, these are not likely to result in substantial changes in the cost differences between the groups.

2. Kindly explain how all relevant cost and effect differences were calculated if [Line 96-97] there is no reliable data available regarding the number of women who undergo specific procedures as a second intervention how can costs be calculated.

We thank the reviewer for this valuable comment. We acknowledge that our explanation in the manuscript may have lacked clarity. We do know exactly which repeat procedures have been performed within the 2 year follow-up of the SAM study, and we have included those costs. There is however no standard protocol on best treatment in cases of recurrence after a sacrospinous hysteropexy or Manchester procedure. We have now clarified this in the paper: “Currently, there is no standard protocol on the best treatment in case of recurrence after a sacrospinous hysteropexy or Manchester procedure.” (lines 94-95)

3. To conclude [line 233-34]that the recurrence rates are lower than other comparative studies 2yr vs 5 year follow up is erroneous.

We thank the reviewer for this comment. The sentence could indeed lead to confusion. We adjusted the sentence for clarity: “In the SAM study, the thresholds for repeat surgery were not reached within two years postoperatively, and it is unlikely that they will be reached at the five-year follow-up. Data for this extended follow-up are currently being collected.” (lines 233-235)

Reviewer #3

The present issue places the consequence of the SAM study, which concluded that sacrospinous hysteropexy is inferior to the Manchester procedure.

Through the present articles, sacrospinous hysteropexy is not entirely inferior to Manchester procedure, taking into consideration about the QALYs.

Actually, almost half of participants belong to the POP-Q 2nd degree. Strictly speaking, those patients tend to be cured even by the use of Pessary, the cheapest procedure. However, this time the highlight is the comparison between sacrospinous hysteropexy and Manchester procedure and not focused on the other procedures.

So far as this second revised articles, there are no suspicious points for me. I really respect your hard works.

Thank you for your thoughtful reflection and kind compliments. We truly appreciate your engagement.

---

## [Decision Letter · Decision Letter 3]

16 Sep 2025

Dear Dr. Schulten;

publication criteria  and not, for example, on novelty or perceived impact.

Please submit your revised manuscript by Oct 31 2025 11:59PM. If you will need more time than this to complete your revisions, please reply to this message or contact the journal office at plosone@plos.org . A rebuttal letter that responds to each point raised by the academic editor and reviewer(s). You should upload this letter as a separate file labeled 'Response to Reviewers'.A marked-up copy of your manuscript that highlights changes made to the original version. You should upload this as a separate file labeled 'Revised Manuscript with Track Changes'.An unmarked version of your revised paper without tracked changes. You should upload this as a separate file labeled 'Manuscript'.

We look forward to receiving your revised manuscript.

Kind regards,

Ozan Karadeniz

Academic Editor

PLOS ONE

Journal Requirements:

Reviewers' comments:

Reviewer's Responses to Questions

**Comments to the Author**

Reviewer #2: All comments have been addressed

2. Is the manuscript technically sound, and do the data support the conclusions?

Reviewer #2: Yes

Reviewer #4: Yes

3. Has the statistical analysis been performed appropriately and rigorously? 

Reviewer #2: Yes

Reviewer #4: Yes

4. Have the authors made all data underlying the findings in their manuscript fully available?

Reviewer #2: Yes

Reviewer #4: Yes

5. Is the manuscript presented in an intelligible fashion and written in standard English?

Reviewer #2: Yes

Reviewer #4: Yes

Reviewer #2: No further comments. All points were clarified and minor corrections made.

I wish the authors all the very best in their work

Reviewer #4: 1. Overall, the study is well-written and flows smoothly. The findings and the authors’ intentions are clear and easy to follow. I appreciate the hard work involved and am impressed by the initial effort that went into designing and completing the SAM study. Based on its results, it is evident that the Manchester procedure is more cost-effective, primarily due to its lower recurrence rate. This naturally may make the readers wonder what the composition of reoperations is, but it is not critical. While the actual costs may not precisely reflect those in other countries (particularly the U.S.), the analysis provides useful context for cost considerations in that setting.

2. The authors have also adequately revised the manuscript in response to the reviewers’ previous comments; however, I have added a few more that need answering.

3. Great spin-off of the initial RCT.

**Do you want your identity to be public for this peer review?**  For information about this choice, including consent withdrawal, please see our Privacy Policy

Reviewer #2: **Yes: ** Lilly Varghese

Reviewer #4: **Yes: ** Youngwu Kim

---

## [Author Response · Author response to Decision Letter 4]

13 Oct 2025

Response to reviewers

Overall comment:

1. Overall, the study is well-written and flows smoothly. The findings and the authors’ intentions are clear and easy to follow. I appreciate the hard work involved and am impressed by the initial effort that went into designing and completing the SAM study. Based on its results, it is evident that the Manchester procedure is more cost-effective, primarily due to its lower recurrence rate. This naturally may make the readers wonder what the composition of reoperations is, but it is not critical. While the actual costs may not precisely reflect those in other countries (particularly the U.S.), the analysis provides useful context for cost considerations in that setting.

2. The authors have also adequately revised the manuscript in response to the reviewers’ previous comments; however, I have added a few more that need answering.

3. Great spin-off of the initial RCT.

Thank you for your thoughtful comments.

Specific comments:

1. Line [63-66] – This statement feels somewhat overstated, even though I recognize it is based on the SAM study results. I suggest using less definitive language than “now that xx is proved to be,” since the Manchester procedure is not currently regarded as the standard of care for all vaginal POP procedures.

The word ‘proved’ has now been amended into ‘appears’. [line 64]

Materials and methods:

1. Clear and well-structured explanation of the materials and methods; acceptable with no major issues noted.

No amendments have been made.

Economic evaluation:

1. The authors’ response to Reviewer #2 regarding the study duration is adequate; however, they could elaborate further than what is currently provided in lines [108–109] to help address potential public concerns when the study is published.

The following sentence has been added with three new references: “These costs include rehabilitation after surgery, management of complications, and treatment of recurrences. This is supported by studies indicating that most recurrences occur within the first two postoperative years.” [line 109-111]

2. Table 1: Why is the re-intervention cost for SSLF/Manchester lower than the primary surgery?

The observed cost differences are attributable to the inclusion of anterior and/or posterior colporrhaphy procedures within the primary surgery costs. As no re-intervention involving sacrospinous hysteropexy was conducted, the corresponding row has been removed from the table. No further amendments were made.

3. Please consider adding a comment on how more recent data, particularly following the COVID-19 pandemic, might influence the findings—whether it would diminish or amplify the cost evaluations based on decade-old data—even if the overall conclusion remains unchanged.

The SAM trial was partly conducted during the pandemic. Many of the procedures were postponed as a result of pandemic related disruptions. However, we do not believe this had a significant impact on the overall costs. Current costs are likely higher due to inflation. However, we like to emphasize that CEA is incremental with regard to analysis by nature, so the difference will relatively be unaffected especially as the randomized clinical trial had a well balanced inclusion regarding control and intervention arm. So both groups met the same COVID-19 challenges. No amendments were made.

4. Additionally, have the authors considered detailing the types of reoperations performed? While not critical, there remains a question as to whether patients in the SSLF group required more “costly” subsequent surgeries. If referenced in the initial SAM study, could reference it.

We thank the reviewer for this suggestion. In table 4 extra costs are mentioned, which include the average cost per participant for repeat surgery (among others). Although the cost difference between the interventions is statistically significant, it constitutes only a minor component of the overall disparity in societal costs. Therefore, we have not detailed the reoperations. The significant difference in societal costs is mainly due to the productivity losses. In the table below, the details of the reoperations are shown.

Sacrospinous hysteropexy

(n=207) Manchester procedure

(n=213)

Repeat surgery for POP 11/207 (5.3%) 4/213 (1.9%)

Repeat surgery operated compartment 9/207 (4.3%) 0

Repeat surgery non-operated compartment 2/154 (1.3%) 4/144 (2.8%)

Type of repeat procedure

Manchester procedure 3 (1.5%) 0

Cervical amputation 2 (1.0%) 0

Posterior colporrhaphy 2 (1.0%) 2 (0.9%)

Anterior colporrhaphy 3 (1.5%) 1 (0.5%)

Vaginal hysterectomy 1 (0.5%) 0

Robot assisted sacropexy 4 (1.9%) 1 (0.5%)

Results:

1. Table 3, Urinary retention with treatments: the numbers of “treatments” do not add up as well as the %.

Table 3 has been adjusted, for clarity another line has been added in the table with the number of women who received both a foley catheter and self-catheterization.

Costs:

- A minor suggestion: consider moving lines [183–185] to follow the first paragraph and place them above Table 4 to improve the overall flow of the study.

The manuscript was amended accordingly.

---

## [Editor Report · Decision Letter 4]

21 Oct 2025

Economic evaluation of Manchester procedure versus sacrospinous hysteropexy: a follow-up analysis of a randomized clinical trial

PONE-D-24-40477R4

Dear Dr. Schulten,

We’re pleased to inform you that your manuscript has been judged scientifically suitable for publication and will be formally accepted for publication once it meets all outstanding technical requirements.

Within one week, you’ll receive an e-mail detailing the required amendments. When these have been addressed, you’ll receive a formal acceptance letter, and your manuscript will be scheduled for publication.

Kind regards,

Ozan Karadeniz

Academic Editor

PLOS ONE

Additional Editor Comments (optional):

The authors are to be commended for conducting a well-designed randomized clinical trial comparing the Manchester procedure and sacrospinous hysteropexy and for providing an important economic evaluation with extended follow-up data. The analyses are clearly presented, and the discussion appropriately contextualizes the findings within the existing literature.

Minor technical issues remain and will be addressed during the final production process. Once these are completed, the manuscript will be ready for publication. The study contributes valuable evidence to guide surgical decision-making in the management of uterine prolapse, especially regarding cost-effectiveness and long-term outcomes.

I would like to sincerely thank all reviewers for their thoughtful and constructive evaluations of this manuscript. Your detailed comments and valuable insights have greatly contributed to improving the scientific quality and clarity of the paper. We truly appreciate the time and expertise you have dedicated to this review process.

---

## [Editor Report · Acceptance letter]

PONE-D-24-40477R4

PLOS ONE

Dear Dr. Schulten,

I'm pleased to inform you that your manuscript has been deemed suitable for publication in PLOS ONE. Congratulations! Your manuscript is now being handed over to our production team.

Kind regards,

on behalf of

MD Ozan Karadeniz

Academic Editor

PLOS ONE